# Estimating GPS-based social aggregation metrics using collar data

William M. Janousek[1]*, Gavin G. Cotterill[1], Olivia J. Lobo[1], Eric K. Cole[2],
Sarah R. Dewey[3], Tabitha A. Graves[1]

**1** U.S. Geological Survey, Northern Rocky Mountain Science Center, West Glacier, Montana, United States of America, **2** U.S. Fish and Wildlife Service, National Elk Refuge, Jackson, Wyoming, United States of America, **3** National Park Service, Grand Teton National Park, Moose, Wyoming, United States of America

* wjanousek@usgs.gov

## Abstract

Understanding social aggregation patterns in ungulate herds is essential for gaining behavioral insights, optimizing resource use, reducing human-wildlife conflict, and managing disease risk. As chronic wasting disease is the preeminent disease-related threat to cervid populations in North America, knowledge of contact between individuals and spatiotemporal patterns of aggregation provides opportunity to understand and potentially reduce disease risk while supporting sustainable population sizes. Herd density metrics, derived from global positioning system (GPS) data, can be used to inform management decisions. To effectively compare aggregation behavior within and between herds, aggregation metrics must be accurate. However, the consistency of metrics across different GPS collar sample sizes remains unclear and robust studies of big game require understanding how these factors may vary in different contexts. We examined the minimum sample size necessary for reliable calculations of three aggregation metrics: pairwise inter-animal distances, daily proximity rates, and kernel density estimate (KDE) areas. We used GPS collar data from the Jackson and West Green River elk herds (*Cervus canadensis*) in western Wyoming, USA, that differ in herd size and group structure (single versus multiple sub-groups), representing common practical contexts. Elk locations were acquired for the Jackson herd between 2016 and 2019 and from 2005 to 2010 for the West Green River herd. Herd-specific characteristics substantially influence the sample size necessary for accurate density measurements. As predicted, larger herds with many groups require more GPS collars than small herds with fewer groups. The sample size needed to accurately estimate aggregation varies by metric, with KDE areas, useful for indexing environmentally transmitted disease risk, generally requiring fewer samples, especially in high-density contexts. The required sample size also varies with seasonal changes in density. During periods of highest density, similar sample sizes are required to estimate inter-animal distances and proximity rates regardless

the Creative Commons CC0 public domain
dedication.

**Data availability statement:** Data availability
statement reads: West Green River elk herd
GPS data are publicly available here: https://
doi.org/10.5066/F70K27SF. Jackson elk herd
GPS data are available upon request via the
National Elk Refuge and Grand Teton National
Park (grte_information@nps.gov, attn: Sarah
Dewey). Raw aggregation metric data used
in this study are available here: https://doi.
org/10.5066/P14KSJQM. Raw GPS-collar data
from elk in the Jackson herd is sensitive due to
concerns about potential misuse, particularly
related to hunting pressure and poaching.
These elk congregate in large numbers within
protected landscapes like national parks and
wildlife refuges during certain times of the year,
making them especially vulnerable if location
data is not handled responsibly. Given these
factors the collar data is available upon request
but continues to be freely shared for research
purposes (recent example: Cotterill et al. 2025
Ecosphere).

**Funding:** Funding was provided by the U.S.
Geological Survey Biothreats Program. The
funders had no role in study design, data
collection and analysis, decision to publish, or
preparation of the manuscript.

**Competing interests:** The authors have
declared that no competing interests exist.

of herd characteristics. Our results have implications for costs associated with studying big game herds, indicating fewer collars may be sufficient in some cases. These insights can aid researchers and managers in determining the appropriate number of GPS collars required for effective herd monitoring and informing relevant aggregation metrics for their management goals.

## Introduction

Wildlife professionals strive to manage big game herds for optimal social, economic and ecological outcomes [1–2]. These interrelated objectives include increasing wildlife-related recreation opportunities that in turn financially support State management agencies while providing local economies with upwards of $400 billion a year in the United States [3]. They also aim to reduce human-wildlife conflict [4], support ecosystem functioning [5], and control the spread of disease [6]. Increasingly, meeting complex management goals requires more sophisticated information, which must be weighed against increased costs of data collection and analysis. Abundance and herd composition estimates, often generated from visual counts, are commonly used to monitor population trends and set population objectives [7]. These estimates are further enhanced by tying them directly to spatial units, and at ecologically relevant scales, because many biological processes exhibit density-dependence [8–9]. At broad spatial scales, population density estimates help managers meet some of the above objectives. For instance, at the level of agency-defined hunt units, density estimates can directly inform sex- and region-specific hunting quotas or indirectly by identifying priority areas for habitat conservation [10]. In contrast, broad-scale density estimates provide little insight into the interactions, individual movements, and 'local densities' experienced by animals that are most salient to disease control [11]. The proliferation of tracking technology has emerged as a valuable tool to meet the need for detailed monitoring of animal movements and density patterns.

Capturing and marking individual animals for research has a long history in wildlife management [12]. Wildlife tracking via satellite was first demonstrated on elk using radio beacon technology and weather satellites [13]. Contemporary global positioning system (GPS) technology now enables the capture of year-round, high-resolution spatial data which has contributed to its adoption and prevalence in wildlife research [14]. Although methods for capturing and handling ungulates have improved, these are still expensive operations (e.g., helicopter flights, weeks of staff time) that can induce stress or risk injuring animal subjects [15–16]. More recently, the addition of proximity sensors to collars has emerged as a promising tool capable of recording pairwise interaction rates among individuals, but these too, require direct capture and handling for deployment [17]. As a result, GPS collars are only ever deployed on a subset of individuals to inform processes representative of the larger population which can create challenges for contact analyses [18]. Nevertheless, GPS collar datasets are ubiquitous and permit the analysis of movement paths, conspecific

contact rates, and spatiotemporal density patterns that inform disease transmission risk [19–20], and in some cases, are retroactively used in epidemiological investigations [21].

As more animals that occupy shared space and have the potential to interact are tracked, GPS data reveal some of the conditions that facilitate transmission of infectious diseases [22–24], including the timing, location, and structure of herd aggregations. Previous works established links between herd aggregations and brucellosis [25–26] and bovine tuberculosis transmission [27]. Chronic wasting disease (CWD), a fatal prion disease spreading rapidly through cervid populations in North America and Fennoscandia [28–29], exhibits characteristics of density dependent transmission [30]. Location data can also be used to predict how a disease would spread if introduced to a novel population by assessing the spatiotemporal patterns of transmission risk [31]. Furthermore, management strategies to reduce aggregation and decrease the potential for disease transmission are being directly informed by GPS location data [32–35]. Even so, the predictions made by any study can only be used effectively if aggregation metrics produced by GPS collar data are accurate.

The topic of sample size requirements has been persistent in conversations about the use of GPS collar data for population-level inference for ecological processes including survival, resource selection, and home range estimation (reviewed by [14]). However, sample size requirements to accurately estimate aggregation patterns of big game herds has received less attention (but refer to [36–38] for studies on other taxa). Multi-agency working groups have identified the management of ungulate density, either through adaptive management protocols or experimental means, as integral to addressing the spread of CWD transmission in North America [39]. Further, targeted reductions in host density through hunting can help manage CWD prevalence [40] and are projected to improve achievement of long-term population objectives [41–42]. As a result, managers and researchers may be interested in devoting some of their resources towards addressing emerging disease threats by better understanding patterns of aggregation in ungulate populations. Informing the knowledge gap of sample size requirements can help direct limited conservation resources including funds and personnel hours.

We focus on three metrics used to understand animal aggregation with implications for direct and indirect pathways of disease transmission: pairwise inter-animal distances, daily proximity rates (a measure of time spent at close distances), and kernel density estimate (KDE) areas [43]. All three metrics are easily calculated using traditional GPS datasets [44] and are loosely analogous to related concepts in social network analysis wherein they describe individual-level, intermediate-level, and group-level measures, respectively [45]. Mathematical disease models traditionally treated transmission processes as frequency- or density-dependent [46], although empirical studies often estimate modes of transmission that fall somewhere between the two extremes [11,47]. Inter-animal distances, proximity rates, and KDE area estimates also follow a frequency-density continuum and thus can inform disease risk. Inter-animal distances (combined with some threshold to index contact probability) align conceptually with contact frequencies, KDEs with densities, and proximity rates are intermediary. Outside disease research, collar-based distance metrics have also been used for analyzing animal social behaviors [38,48] and can form the basis for building formal interaction networks [49].

We investigated the minimum number of GPS collars needed to accurately estimate aggregation metrics using GPS collar data from two elk herds (*Cervus canadensis*) of varying size and structure at two times of the year. Relative ungulate herd densities and contact probabilities fluctuate seasonally (e.g., summer versus winter) with changing biotic and abiotic conditions [33]. During winter, elk herds concentrate on relatively small winter ranges where individuals interact with more conspecifics (homogeneous mixing; Fig 1a). During summer, elk disperse across the landscape which reduces relative densities, and results in fewer conspecific contacts (heterogeneous mixing; Figs 1b,c). We predicted that estimation accuracy of the aggregation metrics would vary by season and reflect different degrees of connectivity among individuals and groups of individuals within a herd. Accurate aggregation metrics should be obtainable at lower sample sizes during times of higher density regardless of absolute herd size. In contrast, during low-density conditions, a large population defined by multiple distinct sub-groups with heterogenous mixing should require a larger minimum sample size

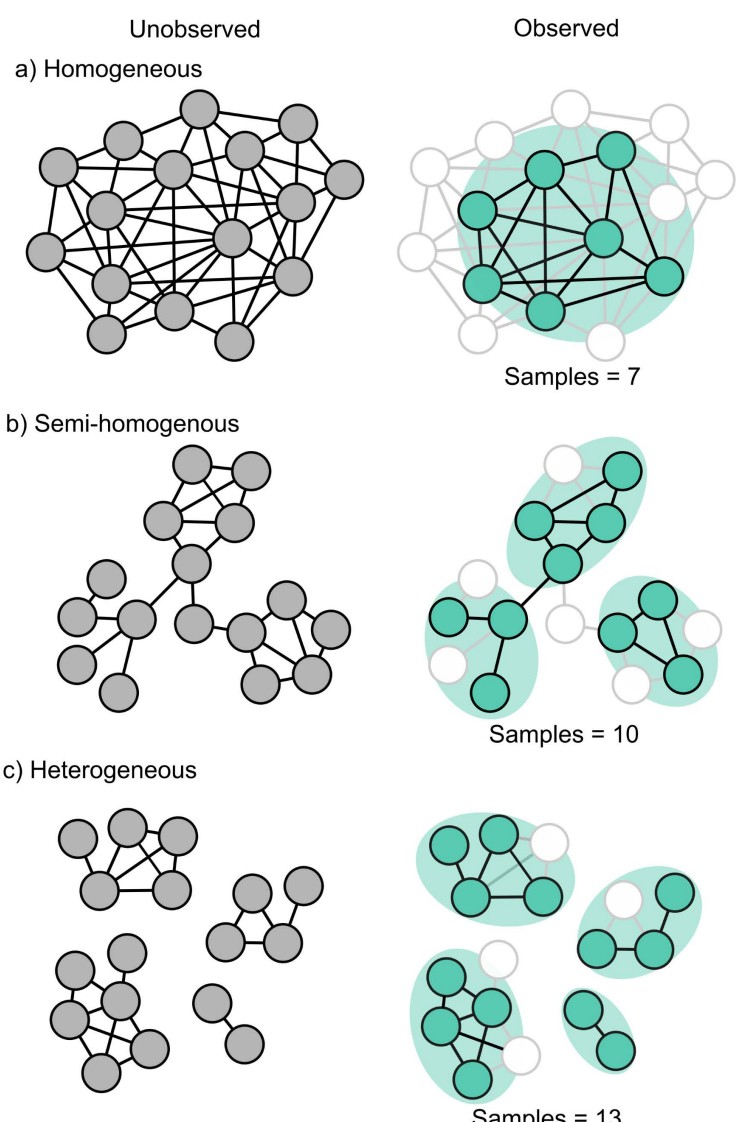

Unobserved  Observed

a) Homogeneous

Samples = 7

b) Semi-homogenous

Samples = 10

c) Heterogeneous

Samples = 13

**Fig 1. Network Structure Concepts.** Diagram showing predictions of variation in network structure within wildlife populations of co-occurring individuals. The full network structure is unobserved without known locations of all individuals but with an adequate sample of the population metrics of aggregation can identify the degree of connectivity. Network structure can range from a single highly mixed homogeneous group (a) to heterogeneous with multiple unconnected groups **(c)**. As network structure changes different numbers of sampled individuals may be necessary for accurate evaluation.

than a small herd that aggregates as a single semi-homogeneous group. We compare the three metrics and discuss how managers can leverage existing and planned GPS datasets to help manage disease risk for their ungulate populations.

## Materials and methods

We investigated aggregation metrics for two elk herds in western Wyoming, USA (Fig 2). Jackson elk (JKSN) were collared on the National Elk Refuge (NER, 43.4805 N, −110.7428 W) and West Green River elk (WGR) were collared on Fossil Butte National Monument (FOBU, 41.8558 N, −110.7615 W). These two elk herds exhibit distinct seasonal movements patterns, leading to differences in their use of seasonal ranges and the extent of intraspecific mixing within each

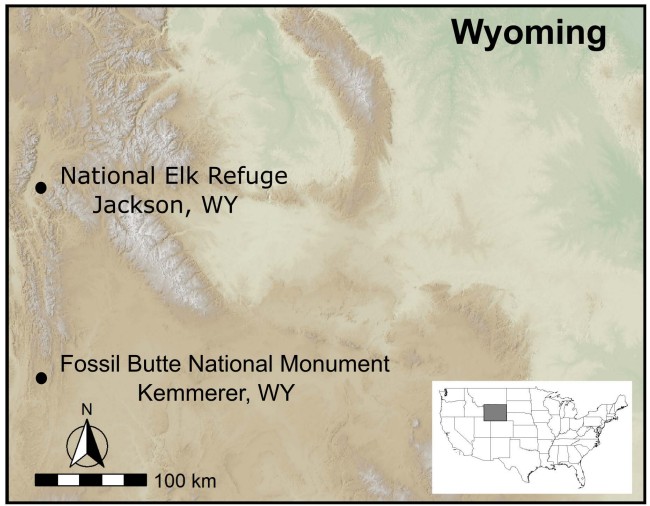

**Fig 2. Study Area Map.** Map showing the study region and locations of the National Elk Refuge and Fossil Butte National Monument in the state of Wyoming, home to the wintering grounds of the Jackson elk herd (JKSN) and West Green River elk herd (WGR), respectively. The background relief is based on the SRTM 30-m digital elevation model provided by NASA JPL [50]. Inset map indicating location of Wyoming derived from **U.**S. Census Bureau boundary data [51].

herd (Fig 3). The JKSN herd inhabit the Middle Rockies ecoregion defined by steep mountains dominated by coniferous forest and shrub- and grass-covered foothills (1,500–3,000 m elevation [52]). During this study, collared elk from the JKSN herd occupied an 1,822 km² area encompassing their wintering grounds on NER and summer range in the Teton and Gros Ventre Ranges, including portions of Grand Teton and Yellowstone National Parks to the west and north and Bridger Teton National Forest to the east (Fig 3a). WGR elk inhabit the foothill shrublands and low conifer-dotted mountains of the Wyoming Basin ecoregion (1,500–2,200 m elevation [52]). Elk from the WGR herd occupied a 730 km² area, including their winter range on Fossil Butte National Monument and summer range in the surrounding Tunp Range which mainly consists of Bureau of Land Management lands (Fig 3c). Both herds experience predation risk from mountain lions (*Puma concolor*), coyotes (*Canis latrans*), black bears (*Ursus americanus*), grizzly bears (*Ursus arctos*), and gray wolves (*Canis lupus*), with the latter two being more prevalent in the JKSN herd range [53].

The JKSN herd has a population of ~11,000 elk [54] and is comprised of disaggregated subgroups (Fig 3a, hetero-geneous mixing) during months of lowest density and a homogenous group during high density winter months (Fig 3b, [55–56]). The WGR herd is estimated to number ~600 elk based on winter herd counts and maintains a consistent homogenous to semi-homogenous group structure (one to few groups) throughout the year (Fig 3c,d, [55–56]). We evaluated GPS collar data with 1.5-hour fix rates for 68 female elk collared between 2016 and 2019 from the JKSN herd after excluding five elk with successful daily fix rates below 0.95 [33]. The WGR herd was monitored with GPS collars recording locations at 5-hour intervals from 2005 to 2010. During that period, 61 female elk were collared for varying lengths of time [55]. Due to declining fix rates over time in the WGR herd, we calculated the rolling seven-day average fix rate for each collar and when the average fell below 0.75, we excluded all subsequent days from our analysis.

We calculated pairwise inter-animal distances, daily proximity rates (a measure of the proportion of the day that two animals are within 500 m of each other), and kernel density estimate areas (KDEs) over the full period available for each herd [33]. In previous work [33], we explored other distance thresholds (100 and 250 m) as alternatives to a 500-m threshold for daily proximity rates and found comparable relationships across methods. At 500 m, we observed the greatest daily fluctuations in proximity rates, which tracked daily changes in abiotic conditions [33]. This indicates that the 500-m

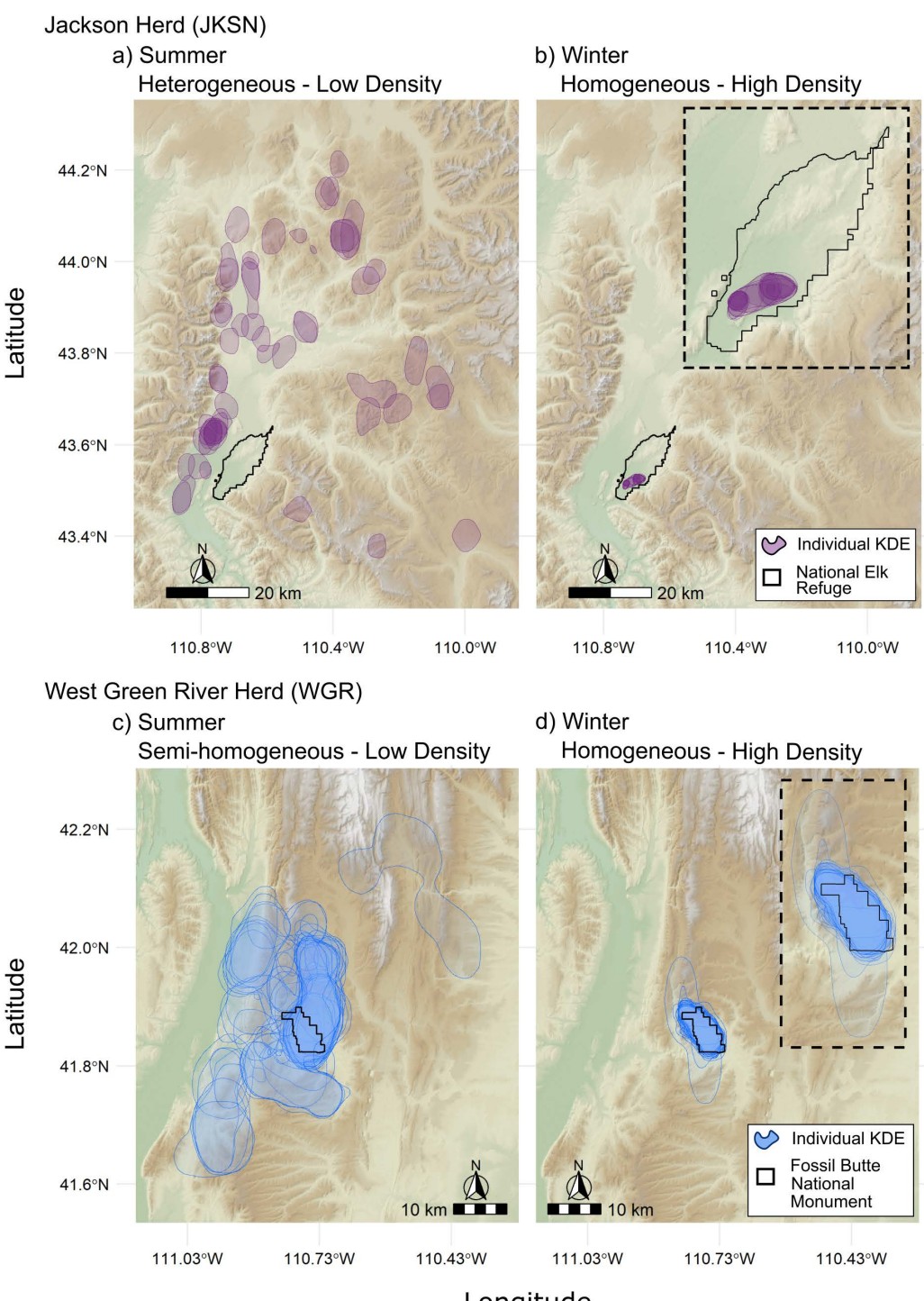

**Fig 3. Seasonal Variation in Herd Structure.** Map showing 90% contour from kernel density estimates for elk from the Jackson (JKSN, purple) and West Green River (WGR, blue) herds during low (a,c) and high density (b,d) periods in Wyoming, USA. The location of the National Elk Refuge and Fossil Butte National Monument are indicated by black bordered polygons. The background relief is based on the SRTM 30-m digital elevation model provided by NASA JPL [50].

threshold is more useful for monitoring proximity rates at fine temporal scales within the context of this study system. Evaluating multiple distances can inform choice of appropriate distance thresholds and may be unique to each study, taxa, spatiotemporal scale, and method or frequency of data collection.

To assess sample size needs, we selected months of lowest and highest densities based on the aggregation metrics. The low- and high-density months included July 2017 (N = 45 elk) and March 2019 (N = 32 elk), respectively for the JKSN herd, and April 2006 (N = 23 elk) and October 2006 (N = 19 elk) for the WGR herd. Because inter-animal distances and proximity rates are daily metrics, we selected the 15th day of each month for our comparison. We summarized pairwise inter-animal distances using the first quartile distance, because this summary is most sensitive to changes in aggregation [43], and we summarized proximity rates using the pairwise mean. We calculated herd-level KDEs and corresponding areas for 50%, 70%, and 90% contours at monthly timescales. We used the default reference bandwidth smoothing parameterization in our KDE estimation, based on the standard deviation of spatial coordinates and sample size [57], to broadly capture the area used by each collared elk population [58]. The smoothing bandwidth can greatly influence how KDEs are drawn, and practitioners can consider their specific ecological context, project goals, and data structure when calculating KDEs. Our interest centered on the effects of changing sample sizes on metric accuracy rather than fine tuning KDEs for more explicit ecological investigation.

For each herd-month-metric combination, we randomly sampled collared elk, incrementing between two and the maximum number of collared elk and resampled 1,000 times with replacement per increment. Assuming the full dataset represented the most accurate value (hereafter baseline value), we assessed variation in aggregation metrics across subsets of collared elk by calculating the percent difference of metrics between the subset and the maximum sample size to determine the accuracy of metrics as sample size increased. For each sample size, we calculated the proportion of subsets within 20% of the baseline value of each aggregation metric. We considered the accuracy threshold to be met when ≥90% of simulations at a given sample size fell within 20% of the baseline value. All aggregation metrics were calculated using the *wildagg* package for program R [44,59].

## Results

The larger JKSN herd typically required larger sample sizes than the smaller WGR herd. Periods when herds were dispersed, with only semi-homogenous and heterogenous mixing (lowest density) required larger minimum sample sizes to achieve accurate measures of all aggregation metrics than periods of homogenous mixing (highest density), regardless of herd identity, (Table 1, Fig 4). During low density months, minimum required sample sizes ranged from 1.9–6.7 times larger than in months of high density for the WGR herd and 1.5–4.2 times larger for the JKSN herd, depending on the aggregation metric (Table 1, Figs 4e-h). To accurately estimate all metrics for the WGR herd, minimum sample sizes of 22 and 16 GPS-collared elk were required under low- and high-density conditions, respectively, which amounts to 96%

**Table 1. Minimum sample size required for aggregation metrics.**

| Metric | WGR | | JKSN | |
| | Low Density | High Density | Low Density | High Density |
|---|---|---|---|---|
| 90% KDE Area | 13 | 3 | 37 | 11 |
| 70% KDE Area | 18 | 3 | 36 | 14 |
| 50% KDE Area | 20 | 3 | 36 | 24 |
| 1st Quartile Inter-Animal Distance | 21 | 16 | 28 | 14 |
| Proximity rate | 21 | 11 | 42 | 10 |

Minimum sample size required to ensure aggregation metrics are within 20% of the baseline value calculated at maximum available sample size for the Jackson (JKSN) and West Green River (WGR) elk herds in Wyoming, USA during periods of low and high density.

Inter-animal Distance

a) High Density

e) Low Density

Proximity Rates

b) High Density

f) Low Density

Kernel Density Estimate - 90% Contour

c) High Density

g) Low Density

Kernel Density Estimate - 50% Contour

d) High Density

h) Low Density

Herd — Jackson — Western Green River

**Fig 4. Sample Size Accumulation Curves for Each Aggregation Metric.** The proportion of simulations at each sample size that are within 20% of the baseline value calculated at maximum available sample size for elk from the Jackson and West Green River herds in Wyoming, USA. Each aggregation

metric is calculated during high- and low-density periods for each herd (a-d, e-h, respectively). Aggregation metrics include the daily 1st quartile inter-elk distance **(a,e)**, mean daily proximity rates **(b,f)**, and the area of 90% (c,g) and 50% contours from kernel density estimates **(d,h)**. Dashed lines indicate when the proportion of simulations within 20% of the baseline value is ≥ 90% (numerical values shown in Table 1). Accumulation curves are fitted using loess smoothing methods for visualization.

and 84% of the maximum available sample size for this herd (Table 1). To accurately estimate all metrics simultaneously for the JKSN herd, sample sizes of 42 and 24 GPS-collared elk were required under low- and high-density conditions, respectively, which reflect 93% and 75% of the maximum available sample size for this herd (Table 1).

Fewer collared elk were needed to produce accurate estimates for KDEs compared to other aggregation metrics for the smaller WGR herd regardless of density conditions (N = 3 under high-density and as few as N = 11 under low-density conditions). In contrast, aggregation metrics requiring the most and fewest collared elk to reach the accuracy threshold were more variable for the JKSN herd. Under high density months, 50% KDE area calculations required the most samples and proximity rates required the fewest samples whereas proximity rates and inter-animal distance metrics required the most and fewest samples during periods of low density for the JKSN herd.

As minimum sample size increases under periods of high density for both elk herds, the proportion of simulations within 20% of the baseline value approach 100% well before reaching the maximum available sample size (Figs 4a-d; but see inter-animal distance estimates for the WGR herd in Fig 4a). In contrast, under low density months, the proportion of simulation estimates within 20% of the baseline value of inter-animal distances, proximity rates, and 50% KDE contour area reach 90% within three sample size increments below the maximum available sample size for the WGR herd (Figs 4e-f and 4h). Proximity rate estimates at low density for the JKSN herd was the only metric to not asymptote at higher sample sizes (Fig 4f).

## Discussion

Herd size, seasonal density patterns and resulting network complexity modulate the minimum sample size required to accurately estimate aggregation metrics for GPS-collared elk. When elk are widely dispersed during the summer months, the overlap of space use between individuals decreases, requiring a larger sample to capture aggregation patterns for the herd. In contrast, when herds are tightly grouped, as they are on wintering grounds, an individual's space use commonly overlaps with others in the herd. The metrics robust to small sample sizes varied both within and between herds, indicating that herd context is an important consideration when determining the optimum sample size and density metrics for studying aggregation behavior in big game populations.

Our results have considerable implications for how managers assess disease transmission risk in ungulate herds. Most pressing in western big game populations is the issue of CWD, which has multiple transmission pathways, adding value to consideration of multiple metrics. Chronic wasting disease can be transmitted by direct contact between individuals or indirectly through the contaminated environment (reviewed by [30]). Assessing risk of direct transmission requires approaches like proximity rates (referred to as contact rates in disease ecology), whereas assessing risk of indirect transmission requires estimating overall space use and defining spatial areas of high concentrations which can be achieved through KDE contours. Our analyses indicate proximity rate calculations need more samples due to this metric requiring GPS-collared animals to be within some minimum distance threshold where interactions occur. For demonstration purposes we used a distance threshold of 500 m in this analysis, which provided robust estimates for understanding the drivers of elk aggregation in previous work [33], but smaller distance thresholds may be required to capture disease-relevant interactions [60]. As distance thresholds decrease, the number of collars in a population will likely need to increase. In contrast, the small sample sizes needed to accurately calculate KDEs likely result from the pooling of location information across individuals to define space use of the herd as a collective. Transmission of any disease is not uniform

  

across space, time, or from individual to individual [61]. Comparisons such as the one we conducted between herds with different sizes and structure can increase our understanding of how transmission risk may differ across unique populations. Our findings that reasonable accuracy for KDE area, proximity rates and inter-animal distance calculations became obtainable for two distinct herds during two periods of different densities and social structure confirm that GPS collar data can provide accurate aggregation metrics to inform decision-making. Studies predict that herd density reduction prior to CWD establishment can result in better long-term outcomes (e.g., higher elk abundance) than actions taken after CWD is established in herds [41–42]. This lends support to multi-agency objectives based on adaptive management approaches targeting ungulate density in various ways to address disease spread [39].

In our study the highest densities and aggregations of elk occur during the winter, when the sexes are together in space and time. In other systems or taxa, the composition of collared individuals (either by sex, age-class, or other characteristics) will need to align with knowledge about transmission dynamics unique to each disease. The mode of transmission (indirect versus direct) will necessarily dictate which aggregations metrics (e.g., KDE for indirect) may be most informative. Researchers should also consider the density characteristics of the population when determining how many collars to deploy and potentially conduct similar resampling analyses as this study presents to fully understand the implications of sampling decisions. This may be especially true for multi-year studies when the opportunity to adjust collar deployments exists. Although our results indicate some aggregation metrics can be accurately measured at low sample sizes, ensuring a representative sample of a herd is integral to avoid sampling individuals that differ considerably in their spatiotemporal movement patterns thereby inducing biased aggregation estimates. Finally, controlling for collar performance as we have done by censoring individuals with low fix rates, can minimize potentially spurious estimates of aggregation.

The aggregation metrics calculated in this study were evaluated within the paradigm of interacting, mobile animals. Fixed points on the landscape, for example attractants like anthropogenic food sources or mineral licks, can also serve as aggregation hotspots increasing the likelihood of direct contact between individuals and between individuals and a pathogen contaminated environment [30,62]. Using the R package *wildagg* [44], as we did to estimate aggregation metrics, managers could treat the locations of attractants as additional 'individuals' to quantify interactions between their wildlife populations and potential point sources of increased pathogen deposition. When the locations of attractants are unknown, cluster analyses may help identify aggregation hotspots [63], which can be described by the aggregation metrics explored in this study.

Managers of big game herds rely on aggregation data to make decisions that can have broad economic and ecological implications such as setting tag quotas, limiting disease transmission risk, or identifying areas with potential for human-wildlife conflict. These results provide researchers and managers with insight into how many GPS collars should be deployed depending on the size and behavior of their herds. With this knowledge, researchers and managers can direct conservation dollars for greatest return on investment.

## Acknowledgments

We thank the Grand Teton Association, U.S. Fish and Wildlife Service, Fossil Butte National Monument, BLM Kemmerer Field Office, Wyoming Game and Fish Department, and the Bridger-Teton National Forest for supporting GPS collar deployment on the National Elk Refuge and Fossil Butte National Monument. Any use of trade, firm or product names is for descriptive purposes only and does not imply endorsement by the U.S. Government.

## Author contributions

**Conceptualization:** William M. Janousek, Olivia J. Lobo, Eric K. Cole, Sarah R. Dewey, Tabitha A. Graves.

**Data curation:** William M. Janousek, Olivia J. Lobo, Eric K. Cole, Sarah R. Dewey, Tabitha A. Graves.

**Formal analysis:** William M. Janousek, Olivia J. Lobo, Tabitha A. Graves.

**Investigation:** William M. Janousek, Gavin G. Cotterill.

**Methodology:** William M. Janousek, Gavin G. Cotterill, Olivia J. Lobo, Sarah R. Dewey.

**Project administration:** Eric K. Cole, Sarah R. Dewey, Tabitha A. Graves.

**Resources:** Eric K. Cole.

**Software:** William M. Janousek.

**Supervision:** William M. Janousek, Tabitha A. Graves.

**Validation:** William M. Janousek, Eric K. Cole.

**Visualization:** William M. Janousek, Gavin G. Cotterill.

**Writing – original draft:** William M. Janousek, Gavin G. Cotterill, Olivia J. Lobo, Tabitha A. Graves.

**Writing – review & editing:** William M. Janousek, Gavin G. Cotterill, Olivia J. Lobo, Eric K. Cole, Sarah R. Dewey, Tabitha A. Graves.

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
