## [Decision Letter · Decision Letter 0]

15 Jul 2025

Dear Dr. Janousek,

Thank you for submitting your manuscript to PLOS ONE. After careful consideration, we feel that it has merit but does not fully meet PLOS ONE’s publication criteria as it currently stands. Therefore, we invite you to submit a revised version of the manuscript that addresses the points raised during the review process.

**While the study addresses an important topic with valuable potential applications, the reviewer has identified critical issues impacting scientific rigor, clarity, generalizability, and practical utility. Addressing these concerns thoroughly—particularly the methodological issues (GPS resolution, sex exclusion, threshold sensitivity), abstract restructuring, seasonal context, disease link clarification, and the addition of a limitations section—is essential for the manuscript to meet publication standards. Major revision is warranted. Acceptance is contingent on satisfactorily addressing all points raised.**

We look forward to receiving your revised manuscript.

Kind regards,

Abebayehu Aticho (PhD, Associate Prof.)

Academic Editor

PLOS ONE

**Journal Requirements:**

1. When submitting your revision, we need you to address these additional requirements. Please ensure that your manuscript meets PLOS ONE's style requirements, including those for file naming. The PLOS ONE style templates can be found at https://journals.plos.org/plosone/s/file?id=wjVg/PLOSOne_formatting_sample_main_body.pdf and https://journals.plos.org/plosone/s/file?id=ba62/PLOSOne_formatting_sample_title_authors_affiliations.pdf 2. Thank you for stating the following financial disclosure: Funding was provided by the U.S. Geological Survey Biothreats Program.   Please state what role the funders took in the study.  If the funders had no role, please state: "The funders had no role in study design, data collection and analysis, decision to publish, or preparation of the manuscript." If this statement is not correct you must amend it as needed. Please include this amended Role of Funder statement in your cover letter; we will change the online submission form on your behalf. 3. We noted in your submission details that a portion of your manuscript may have been presented or published elsewhere. “Some of the raw data used in this analysis has been previously published in a data repository and is cited in text”. Please clarify whether this [conference proceeding or publication] was peer-reviewed and formally published. If this work was previously peer-reviewed and published, in the cover letter please provide the reason that this work does not constitute dual publication and should be included in the current manuscript. 4. In the online submission form, you indicated that your data is available only on request from a third party. Please note that your Data Availability Statement is currently missing the contact details for the third party, such as an email address or a link to where data requests can be made. Please update your statement with the missing information. 5. Thank you for uploading your study's underlying data set. Unfortunately, the repository you have noted in your Data Availability statement does not qualify as an acceptable data repository according to PLOS's standards. At this time, please upload the minimal data set necessary to replicate your study's findings to a stable, public repository (such as figshare or Dryad) and provide us with the relevant URLs, DOIs, or accession numbers that may be used to access these data. For a list of recommended repositories and additional information on PLOS standards for data deposition, please see https://journals.plos.org/plosone/s/recommended-repositories. 6. Please amend the manuscript submission data (via Edit Submission) to include author Dr. Sarah R. Dewey.  7. We note that Figure 2 in your submission contain map images which may be copyrighted. All PLOS content is published under the Creative Commons Attribution License (CC BY 4.0), which means that the manuscript, images, and Supporting Information files will be freely available online, and any third party is permitted to access, download, copy, distribute, and use these materials in any way, even commercially, with proper attribution. For these reasons, we cannot publish previously copyrighted maps or satellite images created using proprietary data, such as Google software (Google Maps, Street View, and Earth). For more information, see our copyright guidelines: http://journals.plos.org/plosone/s/licenses-and-copyright. We require you to either present written permission from the copyright holder to publish these figures specifically under the CC BY 4.0 license, or remove the figures from your submission: a. You may seek permission from the original copyright holder of Figure 2 to publish the content specifically under the CC BY 4.0 license.   We recommend that you contact the original copyright holder with the Content Permission Form (http://journals.plos.org/plosone/s/file?id=7c09/content-permission-form.pdf) and the following text:“I request permission for the open-access journal PLOS ONE to publish XXX under the Creative Commons Attribution License (CCAL) CC BY 4.0 (http://creativecommons.org/licenses/by/4.0/). Please be aware that this license allows unrestricted use and distribution, even commercially, by third parties. Please reply and provide explicit written permission to publish XXX under a CC BY license and complete the attached form.” Please upload the completed Content Permission Form or other proof of granted permissions as an "Other" file with your submission. In the figure caption of the copyrighted figure, please include the following text: “Reprinted from [ref] under a CC BY license, with permission from [name of publisher], original copyright [original copyright year].” b. If you are unable to obtain permission from the original copyright holder to publish these figures under the CC BY 4.0 license or if the copyright holder’s requirements are incompatible with the CC BY 4.0 license, please either i) remove the figure or ii) supply a replacement figure that complies with the CC BY 4.0 license. Please check copyright information on all replacement figures and update the figure caption with source information. If applicable, please specify in the figure caption text when a figure is similar but not identical to the original image and is therefore for illustrative purposes only.The following resources for replacing copyrighted map figures may be helpful: USGS National Map Viewer (public domain): http://viewer.nationalmap.gov/viewer/The Gateway to Astronaut Photography of Earth (public domain): http://eol.jsc.nasa.gov/sseop/clickmap/Maps at the CIA (public domain): https://www.cia.gov/library/publications/the-world-factbook/index.html and https://www.cia.gov/library/publications/cia-maps-publications/index.htmlNASA Earth Observatory (public domain): http://earthobservatory.nasa.gov/Landsat: http://landsat.visibleearth.nasa.gov/USGS EROS (Earth Resources Observatory and Science (EROS) Center) (public domain): http://eros.usgs.gov/#Natural Earth (public domain): http://www.naturalearthdata.com/

**Additional Editor Comments:**

While the study addresses an important topic with valuable potential applications, the reviewer has identified critical issues impacting scientific rigor, clarity, generalizability, and practical utility. Addressing these concerns thoroughly—particularly the methodological issues (GPS resolution, sex exclusion, threshold sensitivity), abstract restructuring, seasonal context, disease link clarification, and the addition of a limitations section—is essential for the manuscript to meet publication standards. Major revision is warranted. Acceptance is contingent on satisfactorily addressing all points raised.

Reviewers' comments:

Reviewer's Responses to Questions

**Comments to the Author**

1. Is the manuscript technically sound, and do the data support the conclusions?

Reviewer #1: Partly

2. Has the statistical analysis been performed appropriately and rigorously?

Reviewer #1: Yes

3. Have the authors made all data underlying the findings in their manuscript fully available?

Reviewer #1: Yes

4. Is the manuscript presented in an intelligible fashion and written in standard English?

Reviewer #1: Yes

**Reviewer #1:**  Line 20 – 24: The general background written doesn’t have continuity with the research statement given. It would be better if the authors start the abstract with a general background on “social aggregation in ungulates, its basic importance (e.g., it helps in shaping behavioral dynamics, and resource use), and how it can lead to infectious disease transmission. This should be followed by introducing the “Ungulate herd density metrics” and why it is important in understanding social aggregation in terms of disease risk and transmission, like chronic wasting disease (CWD). Line 20 – 24: The general background written doesn’t have continuity with the research statement given. It would be better if the authors start the abstract with a general background on “social aggregation in ungulates, its basic importance (e.g., it helps in shaping behavioral dynamics, and resource use), and how it can lead to infectious disease transmission. This should be followed by introducing the “Ungulate herd density metrics” and why it is important in understanding social aggregation in terms of disease risk and transmission, like chronic wasting disease (CWD).

Lines 26-28: This identifies the research gaps, which should be followed by the above statement when revised.

Line 30-32: How many years of data were collected? The study area should be mentioned in the abstract

Line 41-43: In terms of management goals, the author should be specific about what goals we are talking about. Whether it is related to only active management of herds, or any other measures that reduce the disease transmission among populations.

Line 63-64: “The proliferation of tracking…….” can be revised to “The proliferation of tracking technology has emerged as a valuable tool to meet this need by enabling detailed monitoring of animal movements.”

Line 65: Delete “purpose”

Line 67 – 69: Is the reference used here specific to elk or in general?? I suggest adding more recent references specific to elk that justify your statement.

Line 70 – 71: Elaborate on the costly operations that can cause stress or risk to the animal studied.

Lines 72 – 75: For example, proximity sensors are placed on the GPS/GSM/VHF collars that detect proximity between animals or between animals and other species, providing data on interaction rates and contact zones.

Line 85: Explain to the readers how location data can be used to predict disease transmission, and about the tools.

Lines 102-103: The manuscript focuses on the social aggregation of herds and suggests measures to reduce the cost associated with understanding aggregation metrics. Lines 102-103 talk about the cost reduction and personal hours. I suggest writing more information to give a complete picture to the readers, from ecological to economic benefits.

Line 117: The reference cited to support the statement should be across different taxa. Include more references that have studied social behavior among ungulates or other taxa.

Line 118: The comments for Lines 70 – 71 should support this opening statement; that's how this study is important in terms of ecological and economic values.

Line 145 and 146: What are the values inside the parentheses, (NER 43.4805, -110.7428) and (FOBU, 41.8558, -110.7615)?

Line 160: The figure itself is not sufficient to explain the study area. Firstly, I suggest adding the state/country subset map highlighting the study area location. Second, for both the locations, i.e, JKSN and WGR, figure 2 (a) and (c) show the summer conditions while (b) and (d) show winter aggregations. This should appear in the map as summer and winter range, and be included in the figure title as well.

Line 177 - 203:

- There appears to be a potential difference in GPS collar resolution across the two study herds (JKSN and WGR). How do the authors account for fix frequency when comparing daily aggregation metrics like inter-animal distance and proximity rates? Could coarser data bias metric estimation?

-Given that proximity rates and first quartile distances are highly sensitive to fix rate and temporal precision, how accurate are these metrics in periods or individuals with coarser-scale GPS data? Have the authors conducted any sensitivity analysis to evaluate metric performance under lower-resolution scenarios?

-Why were only female elk included in the analyses? Were there male individuals collared during the study period, and if so, what was the rationale for excluding them? The sex-specific social behavior in ungulates could influence aggregation metrics.

-The authors chose a 500 m threshold for defining proximity events and the first quartile of pairwise distances to summarize aggregation. Can the authors justify these thresholds ecologically? Was there any sensitivity analysis performed to explore how changing these values affects aggregation estimates?

-The KDEs were generated using default reference bandwidths without fine-tuning. Could this choice, particularly under varying sample sizes and spatial dispersions, influence the comparability or accuracy of density estimates? This could either over/underestimate the home ranges. If the individuals follow range residency, was any analysis carried out to see their semivariograms? For example, the estimates would vary for home ranges when autocorrelation is accounted for while estimating home ranges.

Line 209-210: The authors report a wide range in minimum required sample sizes (1.9 – 6.7) for WGR and (1.5 – 4.2) JKSN, between low and high density months. Could this variability be an artifact of differences in resolution of GPS fixes between herds or across time? Were collars across sites and years standardized in fix interval, and if not, how might temporal irregularity influence these estimations?

Line 219: The metrics given in the table were tested for robustness under non-random sampling conditions, such as when subsamples included or excluded spatially central vs. peripheral individuals? How might the spatial location of individuals within a herd influence metric estimation bias under small sample sizes?

Line 224: In Figure 3, the manuscript text refers to panels a–h; however, these panel labels are not present in the figure itself. Please revise the figure to include clear and consistent panel labels (e.g., a–h) to match the references in the main text. This will improve clarity and ensure accurate interpretation of the results

Line 242: The proximity rate metric for the JKSN and WGR, herd under low-density conditions, did not asymptote with increasing sample size. Could the authors clarify whether this is due to behavioral heterogeneity (e.g., subgrouping or fission–fusion dynamics) or again a function of resolution or inconsistent temporal overlap among individuals?

Line 251-259: The discussion does not acknowledge the potential effect of irregular GPS collar fix rates or temporal resolutions across sites. Could differences in data resolution, not just herd size or seasonal density, have contributed to variation in required sample sizes? Please clarify if collar performance was controlled for in this interpretation.

Lines 260–273: The discussion draws broad conclusions about disease transmission risk and aggregation behavior without acknowledging the study’s restriction to only female elk. Given that males may exhibit different space use and social structures, how might these sex-based exclusions limit the generalizability of the findings to population-wide disease management?

Lines 270–273: The authors used a 500 m threshold for proximity rates, but note that smaller thresholds may be more relevant for disease transmission. Was any sensitivity analysis performed to evaluate how different thresholds affect sample size needs or proximity estimates? Clarifying this would improve the utility of your findings for applied disease ecology.

Lines 274–276: While KDEs appear robust at low sample sizes, this may be due to the spatial pooling of data. How do the authors reconcile this with the need to capture fine-scale movement or individual-level variation in contact risk?

Lines 288–296: The authors suggest incorporating landscape features like mineral licks as pseudo-individuals for aggregation analysis. How feasible is this with current data, especially if collar resolution or location accuracy is coarse? Can the authors elaborate on how resolution affects the ability to detect point-based aggregation patterns? This should be included in the limitations.

Other comments:

-Seasonal variation in herd dynamics could be introduced in the introduction of the manuscript. Given that herd structure and aggregation patterns vary substantially between summer and winter, and this variation underpins key differences in sample size requirements and aggregation metrics, seasonal dynamics should be introduced in the Introduction. Presenting this context early would help readers better understand the ecological drivers of the study and set the stage for subsequent analyses.

- the seasons and differences in vegetation and landscape structure should be clearly written in the study area section in methods.

-Given the known differences in elk behavior and group structure between summer and winter, more explicit discussion is needed on how seasonal context influences metric sensitivity, contact rates, and implications for disease management.

-Please include a brief discussion of the study's limitations. Specifically, address potential effects of varying GPS collar resolution, the exclusive use of female elk, the fixed 500 m proximity threshold, and the absence of sensitivity analyses for sampling design. Acknowledging these will provide important context for interpreting and applying the results

**Do you want your identity to be public for this peer review?** For information about this choice, including consent withdrawal, please see our For information about this choice, including consent withdrawal, please see our Privacy Policy .

Reviewer #1: **Yes:** Zehidul HussainZehidul Hussain

While revising your submission, please upload your figure files to the Preflight Analysis and Conversion Engine (PACE) digital diagnostic tool, https://pacev2.apexcovantage.com/ . PACE helps ensure that figures meet PLOS requirements. To use PACE, you must first register as a user. Registration is free. Then, login and navigate to the UPLOAD tab, where you will find detailed instructions on how to use the tool. If you encounter any issues or have any questions when using PACE, please email PLOS at . PACE helps ensure that figures meet PLOS requirements. To use PACE, you must first register as a user. Registration is free. Then, login and navigate to the UPLOAD tab, where you will find detailed instructions on how to use the tool. If you encounter any issues or have any questions when using PACE, please email PLOS at figures@plos.org . Please note that Supporting Information files do not need this step.. Please note that Supporting Information files do not need this step.

---

## [Author Response · Author response to Decision Letter 1]

25 Sep 2025

Thank you for the opportunity to revise our manuscript. We have responded to editor and reviewers' comments line by line in the attached documents. Cheers.

---

## [Decision Letter · Decision Letter 1]

17 Nov 2025

Dear Dr. Janousek,

plosone@plos.org . . A rebuttal letter that responds to each point raised by the academic editor and reviewer(s). You should upload this letter as a separate file labeled 'Response to Reviewers'.A marked-up copy of your manuscript that highlights changes made to the original version. You should upload this as a separate file labeled 'Revised Manuscript with Track Changes.'An unmarked version of your revised paper without tracked changes. You should upload this as a separate file labeled 'Manuscript.'

We look forward to receiving your revised manuscript.

Kind regards,

Abebayehu Aticho

Academic Editor

PLOS ONE

Journal Requirements:

**Additional Editor Comments:**

Dear authors, Kindly address all the issues and concerns raised by both reviewers carefully.

Reviewers' comments:

Reviewer's Responses to Questions

**Comments to the Author**

Reviewer #1: All comments have been addressed

Reviewer #2: All comments have been addressed

2. Is the manuscript technically sound, and do the data support the conclusions?

Reviewer #1: Yes

Reviewer #2: Partly

3. Has the statistical analysis been performed appropriately and rigorously?

Reviewer #1: Yes

Reviewer #2: I Don't Know

4. Have the authors made all data underlying the findings in their manuscript fully available?

Reviewer #1: Yes

Reviewer #2: No

5. Is the manuscript presented in an intelligible fashion and written in standard English?

Reviewer #1: Yes

Reviewer #2: No

Reviewer #1: Thank you for addressing all the comments and suggestion. All the reviewers' comments were anserwerd and incorporated in the revised manuscript. There are only a few minor edits that I have commented on in the manuscript file.

Reviewer #2: After a careful review of the paper, I found that the topic pertains to ecology (environmental science), specifically the use of modern technologies to monitor wildlife, behaviors, and populations of animals in a particular area. The paper lacks the technical mechanisms of the Global Positioning System (GPS) and relies solely on data received about the situation in that area. While the research is valuable in environmental science and related fields, and does contribute to those areas, its contribution to the technology itself is insufficient for publication.

GPS data is available for use by researchers in numerous scientific fields, and here the researchers have used this data to monitor and study the behavior of a group of wild animals.

**Do you want your identity to be public for this peer review?** For information about this choice, including consent withdrawal, please see our For information about this choice, including consent withdrawal, please see our Privacy Policy .

Reviewer #1: No

Reviewer #2: No

---

## [Author Response · Author response to Decision Letter 2]

5 Jan 2026

Response to Reviewers

Here we have addressed the reviewer comments found in the ‘Reviewer Questionnaire” email and the line by line responses to comments left in “Reviewer comments” PDF file we received. They are as follows:

Responses to email comments:

Additional Editor Comments: Dear authors, Kindly address all the issues and concerns raised by both reviewers carefully.

Reviewers' comments:

Reviewer's Responses to Questions

Comments to the Author

1. If the authors have adequately addressed your comments raised in a previous round of review and you feel that this manuscript is now acceptable for publication, you may indicate that here to bypass the “Comments to the Author” section, enter your conflict of interest statement in the “Confidential to Editor” section, and submit your "Accept" recommendation.

Reviewer #1: All comments have been addressed

Reviewer #2: All comments have been addressed

Response: We thank the reviewers for their perspectives and time.

2. Is the manuscript technically sound, and do the data support the conclusions?

Reviewer #1: Yes

Reviewer #2: Partly

Response: We have made many of the suggested changes, adding additional context and details that were requested during the first round of review and believe our submission is stronger after the peer review process.

3. Has the statistical analysis been performed appropriately and rigorously?

Reviewer #1: Yes

Reviewer #2: I Don't Know

Response: We believe we have exceeded the standards for statistical rigor in this study.

4. Have the authors made all data underlying the findings in their manuscript fully available?

Reviewer #1: Yes

Reviewer #2: No

Response to Reviewer #2: We have provided contact information for data not readily available due to data sensitivities surrounding information within National Parks. We have published all other underlying data in a static publicly available database (cited in text).

5. Is the manuscript presented in an intelligible fashion and written in standard English?

Reviewer #1: Yes

Reviewer #2: No

We disagree with Reviewer #2’s assessment that the paper is not written in standard English.

6. Review Comments to the Author

Reviewer #1: Thank you for addressing all the comments and suggestion. All the reviewers' comments were anserwerd and incorporated in the revised manuscript. There are only a few minor edits that I have commented on in the manuscript file.

Response: We appreciate the feedback we received and have addressed all minor edits in this last review round.

Reviewer #2: After a careful review of the paper, I found that the topic pertains to ecology (environmental science), specifically the use of modern technologies to monitor wildlife, behaviors, and populations of animals in a particular area. The paper lacks the technical mechanisms of the Global Positioning System (GPS) and relies solely on data received about the situation in that area. While the research is valuable in environmental science and related fields, and does contribute to those areas, its contribution to the technology itself is insufficient for publication.

GPS data is available for use by researchers in numerous scientific fields, and here the researchers have used this data to monitor and study the behavior of a group of wild animals.

Response: The reviewer critiques the paper for not advancing GPS technology, stating that its contribution to technology is insufficient for publication. However, nowhere in the manuscript do we claim to be advancing GPS technology. The study clearly focuses on the ecological application of GPS collar data to investigate aggregation behavior in elk, which we believe is well within the scope of the journal.

Line by line responses:

Line 17. Added the word ‘As’.

Line 18. Deleted the word ‘and’ and added comma per reviewer’s request.

Line 146: Reviewer comment: “I suppose Figure 2 will come here in text.” We imagine this will be an editorial decision on the journal’s part in terms of formatting but yes this paragraph is where Figure 2 is introduced.

Line 151: Reviewer comment: “Consufed about this figure numbers. Kindly please check again.” We have double-checked the figure numbering to ensure they are correct and streamlined where the figures are mentioned in the paragraph in question so that they appear in logical order.

---

## [Decision Letter · Decision Letter 2]

3 Mar 2026

Dear Dr. Janousek,

Thank you for submitting your manuscript to PLOS ONE. After careful consideration, we feel that it has merit but does not fully meet PLOS ONE’s publication criteria as it currently stands. Therefore, we invite you to submit a revised version of the manuscript that addresses the points raised during the review process.

**The paper is substantially improved and almost there, but some relatively minor issues remain to be addressed, as noted by one reviewer.**

We look forward to receiving your revised manuscript.

Kind regards,

Shrisha Rao, Ph.D.

Academic Editor

PLOS One

Journal Requirements:

Additional Editor Comments (if provided):

One reviewer has made some comments and suggestions for improvement. The authors should consider the same to make their final submission.

Reviewers' comments:

Reviewer's Responses to Questions

**Comments to the Author**

Reviewer #1: All comments have been addressed

Reviewer #2: All comments have been addressed

2. Is the manuscript technically sound, and do the data support the conclusions?

Reviewer #1: Yes

Reviewer #2: No

3. Has the statistical analysis been performed appropriately and rigorously?

Reviewer #1: Yes

Reviewer #2: No

4. Have the authors made all data underlying the findings in their manuscript fully available?

Reviewer #1: Yes

Reviewer #2: No

5. Is the manuscript presented in an intelligible fashion and written in standard English?

Reviewer #1: Yes

Reviewer #2: No

Reviewer #1: I have no further comments. All responses have been addressed by the author. All issues are resolved.

Reviewer #2: 1. The Figures are unclear.

2. The research is not in journal template.

3. There is no conclusions section.

4. The research idea is to use GPS and collar data for elk monitoring, etc., but the current research does not provide detailed explanations of this idea.

5. Some abbreviations are undefined.

6. The typesetting and presentation of the research need to be carefully considered.

7. Results are not enough and unclear.

**Do you want your identity to be public for this peer review?** For information about this choice, including consent withdrawal, please see our For information about this choice, including consent withdrawal, please see our Privacy Policy .

Reviewer #1: No

Reviewer #2: No

---

## [Author Response · Author response to Decision Letter 3]

3 Mar 2026

Response to Reviewers

Reviewer #1:

I have no further comments. All responses have been addressed by the author. All issues are resolved.

We thank the reviewer for their time.

Reviewer #2:

1. The Figures are unclear.

2. The research is not in journal template.

3. There is no conclusions section.

4. The research idea is to use GPS and collar data for elk monitoring, etc., but the current research does not provide detailed explanations of this idea.

5. Some abbreviations are undefined.

6. The typesetting and presentation of the research need to be carefully considered.

7. Results are not enough and unclear.

We appreciate Reviewer #2’s efforts. However, the comments provided are broad and do not specify the figures, sections, line numbers, or examples needed to guide targeted revisions. The absence of actionable detail (e.g., “Figures are unclear,” “typesetting needs to be considered,” “results are not enough and unclear”) makes a direct response impractical.

We have re-assessed the journal template and requirements and consider:

1) Figures appear clear (as evidenced by no requests from reviewer #1)

2) Format matches the journal guidelines

3) A conclusions section is not required

4) This is fully described in the Introduction.

5) We reviewed all abbreviations, as did our internal USGS reviewer, to confirm all abbreviations are defined. Some names of data products used in the analysis are acronymically derived and those are cited appropriately.

6) Typesetting is not completed at this stage.

7) Results are clear as evidenced by reviewer #1.

---

## [Editor Report · Decision Letter 3]

9 Mar 2026

Estimating GPS-based social aggregation metrics using collar data

PONE-D-25-26403R3

Dear Dr. Janousek,

We’re pleased to inform you that your manuscript has been judged scientifically suitable for publication and will be formally accepted for publication once it meets all outstanding technical requirements.

Kind regards,

Shrisha Rao, Ph.D.

Academic Editor

PLOS One
---

## [Editor Report · Acceptance letter]

PONE-D-25-26403R3

PLOS One

Dear Dr. Janousek,

I'm pleased to inform you that your manuscript has been deemed suitable for publication in PLOS One. Congratulations! Your manuscript is now being handed over to our production team.

Kind regards,

on behalf of

Dr. Shrisha Rao

Academic Editor

PLOS One